# The Role of Branched-Chain Amino Acid Supplementation in Combination with Locoregional Treatments for Hepatocellular Carcinoma: Systematic Review and Meta-Analysis

**DOI:** 10.3390/cancers15030926

**Published:** 2023-02-01

**Authors:** Georgios A. Sideris, Savvas Tsaramanidis, Aikaterini T. Vyllioti, Njogu Njuguna

**Affiliations:** 1Baystate Medical Center, Department of Radiology, University of Massachusetts Medical School, Springfield, MA 01199, USA; 2Radiology Working Group, Society of Junior Doctors, 11527 Athens, Greece; 3Department of Surgery, Ippokrateio General Hospital of Thessaloniki, Aristotle University of Thessaloniki School of Medicine, 54642 Thessaloniki, Greece; 4Nestlé Clinical Research Unit, 1000 Lausanne, Switzerland

**Keywords:** cancer, hepatocellular carcinoma, liver cancer, cirrhosis, interventional radiology, nutrition, amino acids, supplements

## Abstract

**Simple Summary:**

The use of branched-chain amino acid (BCAA) supplements in patients with cirrhosis and liver cancer has been investigated by numerous studies, with multiple reported benefits including improvements in survival rates and hepatic functional reserve. Although locoregional therapies for liver cancer have gained momentum over the past few decades, the potential role of BCAA supplementation in conjunction with these procedures has not yet been elucidated. In this study, we systematically analyze articles investigating the role of BCAA supplementation in patients with hepatocellular carcinoma undergoing interventional radiology procedures. Our systematic review and meta-analysis reveals that BCAA supplementation is associated with significantly higher post-treatment albumin levels, which may support their use in combination with locoregional treatments for HCC. There is a tendency for improved overall survival, mortality and recurrence rates; however, current data are insufficient to support additional benefits.

**Abstract:**

Background: Branched-chain amino acid (BCAA) supplementation has been linked with favorable outcomes in patients undergoing surgical or palliative treatments for hepatocellular carcinoma (HCC). To date, there has been no systematic review investigating the value of BCAA supplementation in HCC patients undergoing locoregional therapies. Materials and Methods: A systematic search of the literature was performed across five databases/registries using a detailed search algorithm according to the preferred reporting items for systematic reviews and meta-analyses (PRISMA) statement. The search was conducted on 23 March 2022. Results: Sixteen studies with a total of 1594 patients were analyzed. Most patients were male (64.6%) with a mean age of 68.2 ± 4.1 years, Child–Pugh score A (67.9%) and stage II disease (40.0%). Locoregional therapy consisted of radiofrequency ablation, transarterial chemoembolization or hepatic artery infusion chemotherapy. BCAA supplementation was in the form of BCAA granules or BCAA-enriched nutrient. Most studies reported improved albumin levels, non-protein respiratory quotient and quality of life in the BCAA group. Results pertaining to other outcomes including overall survival, recurrence rate, and Child–Pugh score were variable. Meta-analysis showed significantly higher levels of post-treatment serum albumin in the BCAA group (SMD = 0.54, 95% CI 0.20–0.87) but no significant differences in mortality rate (RR = 0.81, 95% CI: 0.65–1.02) and AST (SMD = −0.13, 95% CI: −0.43–0.18). Conclusion: BCAA supplementation is associated with higher post-treatment albumin levels. There are currently not sufficient data to support additional benefits. Further studies are needed to elucidate their value.

## 1. Introduction

End-stage liver disease, also known as cirrhosis, is a major global healthcare concern with significant morbidity and mortality [1]. In the US alone, 4.5 million people suffer from cirrhosis (1.8% of the population), although the prevalence might be underestimated as patients with compensated cirrhosis are frequently asymptomatic [2,3]. Mortality rates related to cirrhosis have been on the rise over the past three decades [4]. It is estimated that up to 3% of cirrhotic patients develop hepatocellular carcinoma (HCC) every year, with an overall 10-year incidence of 29.7% [5]. Hepatitis B and C viruses (HBV and HCV, respectively) are the most common underlying risk factors, increasing the lifetime risk for HCC by up to 20-fold, followed by alcohol abuse and non-alcoholic fatty liver disease (NAFLD) [6].

Amino acid imbalance is one of the hallmarks of chronic liver disease. It is characterized by a decrease in branched-chain amino acids (BCAAs; namely valine, leucine and isoleucine) and an increase in aromatic amino acids (AAAs; namely pheynylalanine, tyrosine and tryptophan). BCAAs are essential amino acids that cannot be synthesized by the body and, therefore, must be obtained by diet, typically vegetables and dairy products. They have multifaceted functions spanning multiple metabolic pathways. They serve as substrates for protein synthesis, increase albumin levels and inhibit proteolysis. Via the IGF-1 pathway, they improve insulin sensitivity, decrease plasma glucose levels and inhibit carcinogenesis. BCAAs play a key role in nitrogen balance as they are a precursor of glutamine, which is crucial for ammonia detoxification. Moreover, they promote liver degeneration, improve immune function and regulate neurotransmission [7,8]. BCAA deficiency can have detrimental consequences such as hypoalbuminemia, hepatic encephalopathy and insulin resistance [7].

More than 90% of HCC cases occur in patients with underlying cirrhosis [9]. Management of HCC depends on tumor extent, liver function and performance status. Approximately two-thirds of patients present with early or intermediate stage, whereas one-third has an advanced stage at diagnosis [10]. Locoregional therapies offered by interventional radiology have been a valuable treatment option for HCC patients, with increasing popularity over the past two decades [9]. Radiofrequency ablation (RFA) is a potentially curative option offered to select patients with early disease, with 5-year overall survival rates up to 70%. Transarterial chemoembolization (TACE) is indicated in patients with intermediate stage disease and may also serve as a bridge to transplantation [11]. Hepatic arterial infusion chemotherapy (HAIC) is offered to patients with advanced disease and involves local infusion of chemotherapeutic medications within the hepatic artery via a pump [12]. Other treatments include radioembolization, microwave ablation and cryoablation. Locoregional therapies can be repeated multiple times to achieve the desired outcome. Their main complication, however, is compromised liver function, especially if repeated sessions are required [13].

Supplementation with BCAA has been proposed as a beneficial adjunct to curative or palliative therapies for HCC. Patients undergoing hepatic resection that received BCAAs were shown to have significant benefits in postoperative morbidity [14]. BCAA supplementation in patients receiving sorafenib helps preserve hepatic function, prevents discontinuation of chemotherapy and improves survival [15,16]. Similar results have been shown in patients undergoing radiotherapy [17]. Several systematic reviews have investigated the role of BCAA supplementation in cirrhotic patients, or in patients undergoing treatment for HCC. Most of the reviews, however, include a combination of therapeutic modalities (i.e., surgical, locoregional and chemotherapy) [18,19]. To date, there has been no systematic review solely focused on locoregional therapies.

The purpose of this study is to systematically review the existing literature related to BCAA supplementation in conjunction with locoregional therapies for cirrhotic patients with HCC and to provide insight on outcomes.

## 2. Materials and Methods

### 2.1. Study Design

A systematic review of the literature was conducted in accordance with the preferred reporting items for systematic reviews and meta-analyses (PRISMA) guidelines [20]. Inclusion criteria were determined by applying the PICO framework:

Participants: patients of any age, sex or race with HCC undergoing minimally invasive therapeutic procedures.

Interventions: BCAA supplementation.

Comparison: no intervention (controls).

Outcomes: overall survival, recurrence rate, complication rate, laboratory values.

Study design: randomized controlled trials (RCT), non-randomized control trials, cohort studies (prospective or retrospective).

Exclusion criteria included: narrative or systematic reviews, meta-analyses, poster abstracts, non-comparative studies, studies involving non-BCAA supplementation, studies not involving locoregional therapies for the treatment of liver cancer, studies involving locoregional and other therapeutic techniques (e.g., resection, conventional chemotherapy) without extractable data for locoregional therapies, studies investigating the effects of supplementation of non-branched-chain amino acids.

### 2.2. Literature Search Strategy

A comprehensive search for eligible studies was performed on 23 March 2022. Overall, four databases (Medline, Cochrane, Embase and Google Scholar) and one registry (Clinicaltrials.gov) were selected. A detailed search algorithm was constructed by a medical librarian, with keywords pertaining to liver malignancies, locoregional treatments and BCAAs. No language or date restrictions were applied. Screening of studies was executed by two investigators (G.A.S. and S.T.). A manual search of the reference lists of each included article was performed to identify missed potentially eligible studies.

### 2.3. Data Collection and Extraction

Two researchers (G.A.S. and S.T.) independently extracted data from all eligible studies. Data tabulation into a standardized Excel spreadsheet was performed.

Extracted variables included study characteristics (institution, type of study and study period), patient demographics (age and gender), baseline clinical characteristics (cause of cirrhosis, cancer stage, alpha-fetoprotein (AFP), Child–Pugh score, tumor size and number and type of interventional procedure), intervention characteristics (type and dose of BCAA supplements, timing of treatment onset and duration of treatment), follow-up time, overall survival (OS), mortality rate, recurrence-free survival (RFS), recurrence rate, event-free survival (EFS), overall event rate, complication rates, laboratory values at end of follow-up (albumin, prothrombin time (PT), total bilirubin (TBil), C-reactive protein (CRP), alanine aminotransferase (ALT), aspartate aminotransferase (AST), branched-chain amino acid-to-tyrosine ratio (BTR), ammonia, cholinesterase (ChE), fasting plasma glucose (FPG), lactate dehydrogenase (LDH), total cholesterol (Tchol), total protein (Tprot), triglycerides (TG), gamma glutamyl transferase (GGT), lymphocyte count (LYMPH), erythrocyte count (RBC), vascular endothelial growth factor (VEGF), homeostatic model assessment for insulin resistance (HOMA-IR), immunereactive insulin (IRI) and non-protein respiratory quotient (npRQ)) and additional outcomes (body mass index (BMI), quality of life scores and sarcopenia indices).

### 2.4. Assessment of Study Quality

Quality assessment was independently performed by two investigators (G.A.S. and S.T.). Discrepancies were resolved by discussion.

The risk of bias in randomized control studies was evaluated using the Cochrane collaboration tool [21]. The tool allows for the assessment of the following bias types: selection bias (random sequence allocation and allocation concealment), performance bias (blinding of participants and personnel), detection bias (blinding of outcome assessment), attrition bias (incomplete outcome data), reporting bias (selective reporting) and other types of bias. Each type of bias is graded based on whether there is a “High”, “Low” or “Unclear” risk of bias in the respective study.

The quality of cohort studies was assessed by the Newcastle–Ottawa scale [22]. The scale uses three benchmarks. The “Selection” benchmark awards four points for each of the following: (1) representativeness of the exposed cohort, (2) selection of the non-exposed cohort, (3) ascertainment of exposure and (4) demonstration that the outcome of interest was not present at the start of the study. The comparability benchmark awards two points for two potential confounders (the factors that were selected were age and Child–Pugh score). The “Outcome” section awards three points for each of the following: (1) outcome assessment, (2) duration of follow-up (only studies with a follow-up time greater than 6 months received a point) and (3) adequacy of follow-up of cohorts. A maximum of nine points can be awarded to each study.

### 2.5. Statistical Analysis

Categorical variables were presented as frequencies and percentages. Continuous variables were presented as means and standard deviations (SD). Comparison of baseline laboratory values between the control and BCAA groups was made through parametric and non-parametric tests based on the type of distribution.

A meta-analysis was performed for selected variables. The difference in continuous variables was expressed as a standardized mean difference (SMD) by dividing the between-group mean difference by the pooled weighted standard deviation (Hedge’s g). Interpretation of the effect estimates was based on Cohen’s d guidelines. The difference in categorical variables was evaluated by calculating the risk ratio (RR). The random-effect method was utilized to aggregate the individual study effects and calculate the summary effect. Statistical heterogeneity was assessed by using Q, tau-square and I^2^ tests. Potential publication bias was evaluated by funnel plots. Statistical significance was set to a *p*-value < 0.05. Data analysis was performed with R Studio (version 4.1.0, 2021, R Statistical Software Foundation for Statistical Computing, Vienna, Austria).

## 3. Results

### 3.1. Study Characteristics

Our comprehensive literature search yielded 346 results, of which 16 met inclusion criteria [23,24,25,26,27,28,29,30,31,32,33,34,35,36,37,38]. A PRISMA flowchart (Figure 1) was generated using an online tool [39]. No pertinent active clinical trial was identified.

Details about the included studies and patient demographics are presented in Table 1. Study periods ranged from 1998 to 2014, and publication dates ranged from 2004 to 2016. Six of the included studies were randomized controlled trials (RCTs) (one was a pilot study), and ten were cohort studies (seven were retrospective and three were prospective). The country of origin was Japan in fifteen studies and Hong Kong in one. Two studies were conducted in the same institution but had no overlapping study dates, so both were included [24,26]. Two studies were conducted in the same institution and had overlapping dates but examined different outcomes, so both were included [30,31]. Two studies were conducted in the same institution with uncertain overlap (as one of the two did not report on the study period) but examined different outcomes, so both were included [37,38].

### 3.2. Quality of Evidence Assessment

The six RCTs were assessed with the Cochrane tool (Figure 2). All studies had a high risk of performance bias and a high or unclear risk of detection bias. Risk for other types of bias were overall low.

The 10 cohort studies were assessed with the Newcastle–Ottawa scale (Figure 3). Mean score was 7.7, indicating high overall quality. All studies received maximum points in the “Selection” section. Eight out of the ten studies were awarded only one point in the comparability section, as they did not compare the two groups by Child–Pugh score. Five out of ten studies missed a point on the “Outcome” section because of short follow-up time.

### 3.3. Baseline Patient Characteristics

The eligible studies included a total of 1594 patients, 915 of whom were controls and 679 of whom received BCAA supplementation (BCAA group). Of the total, 1022 (64.6%) patients were men. Mean age in the control and BCAA groups was 68.4 ± 4.3 and 68.0 ± 4.0, respectively (*p* < 0.05).

The underlying cause of cirrhosis was mentioned in 12 out of 16 studies. HBV was present in 288 patients (25.2%), HCV in 687 (60.2%) and other in 167 (14.6%). All but four studies reported the Child–Pugh score at baseline, which was A in 767 patients (67.9%), B in 337 (29.8%) and C in 25 (2.2%). Tumor stage at baseline was provided by 12 studies, all of which used the Liver Cancer Study Group of Japan staging system [40]. Tumor stage was I in 405 patients (28.7%), II in 565 (40.0%), III in 284 (20.1%) and IV in 157 (11.1%). Mean maximum tumor diameter was 2.7 ± 1.5 cm in the control group and 2.7 ± 1.5 cm in the BCAA group (*p* < 0.05). Mean number of tumors per patient was 1.5 ± 0.4 in the control group and 1.5 ± 0.4 in the BCAA group (*p* < 0.05).

### 3.4. Baseline Laboratory Values

Mean values of the laboratory markers for each group are shown in Table 2. There were no significant differences in the measured variables between the two groups.

### 3.5. Locoregional Treatments

With regards to the primary liver cancer intervention, RFA was used in ten studies [24,25,26,28,29,31,34,36,37,38], TACE was used in five [24,26,30,33,35] and HAIC was used in three [23,26,27] (Table 3). One study had two separate cohorts with RFA and TACE [24]. Overall, 941 (59.0%) patients underwent RFA, 458 (28.7%) patients underwent TACE, 76 (4.8%) underwent both RFA and TACE and 119 patients (7.5%) underwent HAIC.

### 3.6. BCAA Supplements

BCAA supplementation consisted of BCAA granules (LIVACT granules, Ajinomoto Pharmaceuticals, Tokyo Japan) in 7 studies [24,30,31,34,36,37,38] and of BCAA-enriched nutrient (Aminoleban EN, Otsuka Pharmaceuticals, Tokyo, Japan) in 7 studies [23,25,28,29,32,33,35]. Two studies used both [26,27] (Table 3).

The dose of LIVACT granules was 4.15 g three times a day in all studies. The dose of Aminoleban EN was 50 g once a day in 3 studies [23,29,35], and twice a day in 5 studies [26,27,28,32,33]. The dose of Aminoleban EN was not reported in 1 study [25]. In 2 of the 3 studies providing Aminoleban EN in a single daily dose, supplementation was given as a late evening snack at 10 p.m. [23,35]. In 1 study, the BCAA cohort was equally divided into 2 subgroups: morning (7–10 a.m.) and late evening (10 p.m.) Aminoleban supplementation [29].

The timing of onset of supplementation was not reported in 9 out of 16 studies. Supplementation was initiated between 12 weeks and 1 day prior to the locoregional therapy in 6 studies [24,26,32,33,34,35], whereas in 1 study the onset was immediately after treatment [27].

Only 4 studies discussed the daily calory and protein intake of their subjects and reported no significant differences between the 2 groups [23,28,32,34].

### 3.7. Follow-Up Time

Follow-up time ranged from 10 days to 60 months. In 4 out of 16 studies, the patients were followed up for 3 months or less [24,26,29,34].

### 3.8. Overall Survival (OS)

OS was reported in six studies for a total of 737 patients.

Two studies found improved OS in the BCAA group. In the study by Nishikawa et al., the 1-, 3- and 5-year OS rates in the BCAA group (94%, 70% and 39%, respectively) were significantly higher than in the control group (94%, 49.8% and 21.2%, respectively). On multivariate analysis, BCAA treatment and serum albumin level of ≥3.4 g/dL were significant independent risk factors, with hazard ratios and 95% CI of 1.67 (1.15–2.42) and 1.26 (0.98–1.53), respectively. Subgroup analysis demonstrated that patients with diabetes mellitus and with BMI ≥25 kg/m^2^ had significantly decreased OS rates [31]. In the study by Nojiri et al., OS rates at 1, 3 and 5 years were significantly higher in the BCAA group (100%, 100% and 66%, respectively) than in the control group (100%, 92% and 18%, respectively) [32].

Three studies showed improved OS in the BCAA group only in certain subgroups. Harima et al. found no significant differences in OS between the two groups. However, among patients with stable or progressive disease, there was significantly prolonged survival in the BCAA group [23]. Kanekawa et al. found significantly longer median survival time in Child–Pugh B patients in the BCAA group, but not in Child–Pugh A patients [27]. Tsuchiya et al. showed that OS was significantly longer in the BCAA group in patients with baseline albumin <3.5 mg/dL (despite lower baseline hepatic reserve compared to controls) but not in the group with >3.5. No significant differences in OS were found between the two groups when the patients were stratified by BMI [36].

One study showed no significant differences in median OS in the controls (905 days) compared to the BCAA group (935 days) after a mean follow-up of 2.7–2.9 years [35].

### 3.9. Mortality Rate

Overall, five studies with a total of 449 patients reported mortality rates. Three out of five studies found no significant difference in mortality rates between the two groups after a follow-up period of 12–30 months [23,28,33]. Two studies demonstrated significantly higher mortality rates in the control group. Mortality rates in the control and BCAA groups respectively were 59.6% versus 40.9% after a mean follow-up of 2.5–2.7 years in one study [30], and 38.5% versus 12% after a follow-up of 60 months in the other study [32].

A meta-analysis of these five studies was performed (Figure 4). There was no statistically significant difference in mortality rates between the two groups (RR = 0.81, 95% CI: 0.65–1.02), with a tendency for lower rates in the BCAA group. Heterogeneity among studies was moderately high (I^2^ = 45.2%).

### 3.10. Recurrence-Free Survival (RFS)

Overall, two studies with a total of 155 patients reported RFS. One study showed significantly improved RFS in the BCAA group. The 1-, 3- and 5-year RFS rates were 52%, 12% and 5.2%, respectively, in the control group, and 61.8%, 28% and 16.8%, respectively, in the BCAA group (*p* = 0.013). On multivariate analysis, age greater than 70 years, BCAA supplementation and serum albumin level of ≥3.4 mg/dL were significant independent risk factors, with hazard ratios and 95% CI of 0.70 (0.50–0.98), 1.5 (1.1–2.0) and 1.37 (0.93–1.87), respectively. Subgroup analysis demonstrated no significant differences in patients with diabetes mellitus or BMI ≥25 kg/m^2^ [31]. One study showed no significant differences in median RFS in the control group (358 days) versus the BCAA group (385 days) after a mean follow-up time of 2.7–2.9 years [35].

### 3.11. Recurrence Rate

Overall, six studies with a total of 558 patients reported recurrence rates. Three studies showed no significant differences in recurrence rates among the two groups after a follow-up time ranging between 12 and 40 months [28,36,37]. Three studies showed significantly lower recurrence rates in the BCAA group. In the study by Nishikawa et al., the recurrence rate was 74.5% in the control group versus 60% in the BCAA group during a mean follow-up period of 2.5 years [31]. In the study by Nojiri et al., cumulative recurrence rates at 1, 3 and 5 years were significantly lower in the BCAA group (12%, 44% and 58%, respectively) than in the control group (12%, 68% and 93%, respectively) [32]. In the study by Yoshiji et al., when patients were stratified by baseline HOMA-IR of >2.5 or by fasting insulin (IRI > 15 U/mL), the BCAA group had lower recurrence rates [38].

### 3.12. Event-Free Survival (EFS)

Two studies with a total of 86 patients reported EFS. In one study, cumulative EFS at 1-, 3-, and 5 years was significantly higher in the BCAA group (96%, 74% and 40%, respectively) than in the control group (92%, 65% and 10%, respectively) [32]. In the other study, there was a tendency for longer EFS in the BCAA group (100% and 90% at 6 and 12 months, respectively) than in the control group (79% and 73% at 6 and 12 months, respectively), but without statistical significance [28].

### 3.13. Overall Events/Complications

Three studies with a total of 170 patients reported overall event rates. In two studies, the overall event rate was significantly lower in the BCAA group. In the study by Nojiri et al., the overall event rate after a follow-up of 5 years was 69.2% in the control group versus 40% in the BCAA group [32]. Similar findings were shown in the study by Poon et al. after a 1-year follow-up (37.2% in the control group versus 17.1% in the BCAA group) [33]. One study showed no significant difference between the two groups at 12 months (26.7% in controls versus 10% in the BCAA group) [28].

The frequency of ascites was significantly higher in the control group in two studies [32,33] and not significantly different in one study [31]. Peripheral edema was significantly more common in the control group in one study [33]. No significant differences between the two groups were shown in the rates of liver failure [28], variceal rupture [28,32,33], jaundice [32], hepatic encephalopathy (HE) [32,33], tumor rupture [33], liver abscess [31,33], biloma [31], intra-abdominal bleeding [31], renal failure [33] and pneumothorax/hemothorax [31]. One study showed that readmission rates for complications were significantly lower in the BCAA group during the 12-month follow-up [33].

Only one study reported supplement-induced side effects [29]. In this study, two patients developed hyperglycemia (both of which were diabetics), and one experienced supplement-induced vomiting.

### 3.14. Child-Pugh Score

Overall, five studies with a total of 469 patients reported post-treatment changes in the Child–Pugh score. Three studies demonstrated significant differences in Child–Pugh scores between the two groups at the end of follow-up. In the study by Morihara et al., the Child–Pugh score was significantly improved at 12 weeks in the nocturnal BCAA supplementation group. Late evening snack was independently associated with lower Child–Pugh scores on multivariate analysis (OR 12.5, 95% CI 1.20–152.21) [29]. In the study by Nishikawa et al., the BCAA group had overall significantly improved Child–Pugh scores at 3 and 6 months. A subgroup analysis was performed. Among patients with a baseline Child–Pugh score of A, the BCAA group had significantly lower Child–Pugh scores at 3 and 6 months. Among patients with a baseline Child–Pugh score of B, the BCAA group had significantly lower Child–Pugh scores at 1, 3 and 6 months. The BCAA group was also associated with significantly lower Child–Pugh scores at 3 and 6 months both in the high or low epirubicin groups [30]. Another study showed significantly improved Child–Pugh scores in the BCAA group among patients with a baseline Child–Pugh score of B but not in those with a baseline score of A [27]. Two studies demonstrated no significant changes in Child–Pugh scores at 2–12 months [32,34].

### 3.15. Albumin

Overall, 13 studies with a total of 1127 patients reported post-treatment changes in serum albumin levels; 11 out of 13 studies reported favorable outcomes in the BCAA group. Significantly improved albumin levels were shown in the BCAA group after a follow-up ranging between 0.5–12 months [23,27,28,29,30,32,33,34]. A study performing subgroup analysis demonstrated significantly higher albumin levels with BCAA supplementation regardless of the baseline Child–Pugh score or epirubicin dose at 1, 3 or 6 months [30]. One other study showed a greater increase from baseline in the nocturnal supplementation group compared to the morning supplementation and control groups [29]. There was a significantly lower decrease from baseline in the BCAA group compared to the controls in three studies at 2–4 weeks, which was independent of the Child–Pugh score [24,26,35]. Two studies reported no significant differences at 1, 6 or 12 months post-treatment [25,37].

Four studies with a total of 212 patients were included in the meta-analysis [28,32,33,37] (Figure 5). There were no significant differences in the baseline albumin levels in the included studies. Post-treatment albumin values were significantly greater in the BCAA group with a medium effect (SMD = 0.54, 95% CI 0.20–0.87). Heterogeneity among studies was low (I^2^ = 28.64%). No publication bias was present.

### 3.16. AST and ALT

Overall, nine studies with a total of 703 patients reported differences in AST or ALT from baseline after treatment. Eight out of nine studies demonstrated no significant changes among the two groups throughout a follow-up period ranging from 7 days to 60 months [24,27,28,32,33,34,35,37]. One study showed significantly improved ALT levels in the BCAA group at 5 weeks [23].

Three studies with a total of 177 patients reporting on AST levels were included in the meta-analysis [32,33,37] (Figure 6). All three studies had a follow-up of 12 months. There were no significant differences in the baseline AST levels in the included studies. Post-intervention meta-analysis showed no significant differences between the two groups (SMD = −0.13, 95% CI: −0.43–0.18). Heterogeneity among studies was low (I^2^ = 0.0%). No publication bias was present.

### 3.17. Ammonia

Two studies with a total of 176 patients reported ammonia changes post-treatment. There was a significantly higher decrease in the BCAA group at 2–4 weeks after the onset of treatment compared to controls [26,35]. In one of the two studies, the difference was significant only among Child–Pugh A patients [26].

### 3.18. TBil

Overall, six studies with a total of 337 patients reported the changes in TBil from baseline. Four studies demonstrated no significant differences between the two groups at 0.5, 3 or 12 months [27,28,34,35]. Two studies showed significantly lower TBil levels in the BCAA group at 3 and 6 months [29,33]. Significance was only shown in the late evening snack group in one of these studies [29].

### 3.19. PT

Overall, six studies with a total of 332 patients reported changes of PT from baseline. Five out of the six studies showed no significant differences among the two groups at 12 months of follow-up [27,28,29,32,33]. One study showed higher PT levels in the BCAA group after 3 months [34].

### 3.20. BTR

Overall, six studies with a total of 555 patients reported changes of BTR from baseline [23,24,26,28,32,35]. Four out of the six showed no significant differences between the two groups at 0.5–12 months [24,26,28,35]. Two studies showed improved BTR in the BCAA group [23,32].

### 3.21. Other Laboratory Values

One study showed a significantly lower elevation of CRP from baseline in the BCAA group 2, 5 and 10 days after TACE and 2 and 5 days after RFA, independent of the baseline Child–Pugh score [24]. Another study showed no significant changes at 7 days [34]. One study showed increased VEGF levels at 12 months in the control group [38].

One study showed a significantly greater decrease in total protein and total cholesterol in the control group than in the BCAA group at 2 weeks [35]. Two studies showed greater decreases of ChE from baseline in the control group at 2–5 weeks, despite higher baseline levels in the control group [23,35]. No difference was shown at 3 months in one study [34]. No statistically significant differences were found among the two groups with regards to changes from baseline of FPG [23,28,32,35,37], HOMA-IR [23,37], GGT [34,35], LDH [34,35], TG [35], IRI [23,35] and BMI [23,33,35].

One study showed a greater decrease in erythrocyte count in the control group than the BCAA group at 2 weeks, without any significant changes in leukocyte or platelet counts [35]. One study showed increased lymphocyte counts in the BCAA group at 12 months [32], whereas another study showed no change at 7 days [34].

Three studies with a total of 98 patients showed significant increases in npRQ in the BCAA group at 7 days, 5 weeks and 3 months after treatment [23,28,34].

### 3.22. Quality of Life and Sarcopenia Indices

Two studies showed significantly improved SF-8 scores in the BCAA group at 12 months compared to the control group, including general health perception, physical, emotional and social functioning and mental health [28,32]. One study showed a higher FACT-G score in the BCAA group at 12 months [33].

One study showed a significantly better hand grip strength in the BCAA group at 3 and 6 months [33]. No significant changes in mid-arm circumference, triceps skin fold, skeletal muscle mass, body fat mass and fat-free mass were shown in two studies at 5 weeks–12 months [23,33].

A summary of evidence for all variables is presented in Table 4.

## 4. Discussion

Cirrhotic patients suffer from protein-energy malnutrition, characterized by accelerated gluconeogenesis, proteolysis and lipolysis, resulting in the depletion of albumin, adipose tissue and muscle mass [41,42]. Malnutrition is further propagated by gut dysmotility, malabsorption and anorexia related to portal hypertension. In such hypercatabolic states, BCAAs have higher energy efficiency than glucose or fatty acids and are therefore highly desired energy substrates. They are also utilized in skeletal muscle for glutamate synthesis, which is needed for ammonia detoxification. Due to increased demand, their levels decrease in cirrhotic patients despite adequate dietary intake [42]. AAA levels in turn increase as a result of decreased hepatic uptake and portosystemic shunting. The Fischer ratio (BCAA: AAA) and the BCAA-to-tyrosine ratio (BTR) are useful indices reflecting derangements in amino acid concentrations. They also have a prognostic role as they precede changes in albumin levels and can predict the progression of liver dysfunction [43]. Although the Child–Pugh score is the most widely used assessment method for hepatic functional reserves with excellent prognostic value, it does not include the status of amino acid metabolism.

BCAA supplementation has been included in the recommendations of various nutrition and hepatology organizations worldwide. The 2002 guidelines of the American Society for Parenteral and Enteral Nutrition (ASPEN) recommend BCAA-enriched supplements only in cirrhotic patients with chronic hepatic encephalopathy refractory to pharmacotherapy (Grade B recommendation) [44]. The 2012 Japanese Nutritional Study Group recommends the administration of BCAA to liver cirrhotic patients who have serum albumin levels of 3.5 g/dL or less, a Fisher ratio of 1.8 or less and a BCAA-to-tyrosine ratio of 3.5 or less [45]. The 2019 guidelines by the European Society for Clinical Nutrition and Metabolism (ESPEN) recommend the use of vegetable proteins or BCAA (0.25 g/kg per day) in meat-protein-intolerant cirrhotic patients (Grade B recommendation). They also recommend long-term oral BCAA supplements (0.25 g/kg per day) in patients with advanced cirrhosis to improve EFS and QoL (Grade B recommendation) [46]. The 2019 European Association for the Study of the Liver (EASL) guidelines also recommend the use of BCAA supplements in decompensated cirrhotic patients (strong recommendation, low-quality evidence), and in patients with HE (strong recommendation, high-quality evidence) [47]. BCAA supplementation is generally not recommended in acute liver failure, which is characterized by increased BCAA levels caused by leakage from the damaged hepatocytes [48].

There are currently no guidelines for the appropriate timing of supplementation, or for the duration of treatment. Most of our included studies initiated BCAA therapy prior to the onset of the locoregional procedures (ranging from 1 day to 4 weeks prior). Nocturnal BCAA supplementation has been greatly favored in the literature over daytime supplementation. Cirrhotic patients enter an accelerated state of starvation during nocturnal hours that is equivalent to the starvation experienced by healthy patients after a 2–3 day fast [49]. This is characterized by rapid protein breakdown, consumption of amino acids and an increase in gluconeogenesis. The overnight hypercatabolic state can be mitigated by a late evening snack. BCAAs are the supplement of choice as carbohydrates would lead to further hyperglycemia [50]. BCAAs have a greater capacity to increase albumin synthesis at nighttime rather than during diurnal hours when they mostly serve as energy substrates. A late evening snack is also associated with increased muscle strength and serum albumin compared to daytime administration [51,52]. One of the included studies performed a comparison between early morning and late evening BCAA supplementation in conjunction with RFA showed significantly improved Child–Pugh score, albumin and TBil in the late-evening group; however, the sample size was small and follow-up time was 12 weeks [29].

There is a lack of consistency in the literature with regard to the relative proportions of the administered BCAAs, frequency of administration and total BCAA dose. Our studies included two forms of supplements: BCAA granules and BCAA-enriched nutrients. In total, 50 g of Aminoleban EN consist of valine (1.6 g), leucine (2.0 g) and isoleucine (1.9 g), and from 6.5 g of free amino acids. BCAA granules contain L-isoleucine (0.9 g), L-leucine (1.9 g) and L-valine (1.1 g) per sachet and are administered at a dose of one sachet three times a day. The small granules reduce the stimulation of taste buds and may have improved compliance. Our studies generally aimed at doubling dietary intake by providing total BCAA doses of 0.25 g/kg/d, which exceed the proposed requirements for healthy adults (0.145 g/kg). This is in line with findings from other meta-analyses [52]. Leucine is the most potent BCAA and could potentially be administered alone, achieving similar effects at smaller doses [42]. However, administering leucine alone may have negative consequences as the decreased availability of valine and isoleucine may interfere with protein synthesis and thus reduce the overall treatment efficacy [53].

Based on our systematic review, there is no robust evidence to suggest that BCAA supplementation can provide better outcomes with respect to OS, mortality rate, recurrence rate and RFS. Furthermore, our meta-analysis showed no significant benefits in mortality rates in the BCAA group. One of the included studies showed longer OS in Child–Pugh B patients [27]. Our findings are in line with another meta-analysis that included 974 HCC patients receiving surgery, locoregional therapy or chemotherapy. It showed significantly improved mortality in Child–Pugh B patients with HCC receiving BCAA compared to Child–Pugh A patients [18]. It is postulated that patients with Child–Pugh B benefit the most from BCAA supplementation and exhibit a more rigorous response because of low baseline albumin levels. Recurrence rates and 1-year mortality were not different in the two groups, but 3-year mortality was significantly lower in the BCAA group in this study. Another meta-analysis with a total of 1179 patients undergoing surgery and interventional procedures for HCC showed that BCAAs had no significant effect on 1-year mortality but were associated with significantly lower mortality at 3 and 5 years after treatment [19]. This may suggest that more than 1 year of supplementation may be required for mortality benefits.

Following locoregional therapies, there is a transient decrease in albumin levels caused by local hepatic injury. This is attributed to decreased hepatocyte count, inflammatory cytokines and leakage of albumin because of inflammation of treated areas. It occurs as early as 3 days post-procedure, with most patients recovering within a month. BCAAs can shorten the recovery of post-operative albumin drop, returning to baseline levels by day 10 [24]. Patients with Child–Pugh scores of B or C have fewer hepatic functional reserves and, therefore, may take longer to recover [54,55]. Low albumin levels secondary to the impaired synthetic capacity of the liver or protein malnutrition have a significant prognostic role in patients with cirrhosis and are associated with negative outcomes.

Based on our review, there is evidence to suggest that BCAA supplementation can help prevent the post-treatment drop of albumin and maintain high albumin levels. Eleven out of the 13 included studies that examined the changes of serum albumin demonstrated significant improvements from baseline in the BCAA group. This was further corroborated by our meta-analysis of four of these studies. A possible explanation for the lack of significant changes from baseline in some of the studies in the literature is that BCAAs not only enhance albumin production but also improve albumin quality. BCAAs increase the reduced oxidized albumin ratio, even in the absence of measurable changes in the levels of total albumin, leading to lower rates of ascites [41,43,56].

BCAA supplements are generally more effective in severely catabolic patients with a lower Fischer ratio. However, several studies have suggested that BCAA supplementation in patients with compensated cirrhosis and albumin levels greater than 3.6 g/dL or with a BTR of 4 or less may prevent a future drop in albumin [30,57]. In a cohort of 1134 patients with normal albumin at baseline undergoing surgical, locoregional or chemotherapy treatments for HCC, it was shown that patients taking BCAA with albumin 3.6–4.0 and with Stage III/IV disease had a longer OS compared to controls with the same stage and albumin levels. The authors suggested that early BCAA therapy (during the compensated phase of cirrhosis) may be beneficial for select patients. No differences in OS were shown in patients with albumin greater than 4.0 g/dL, suggesting that very early therapy may not be necessary [58].

BCAAs have been shown to improve hepatic functional reserve and decrease the rate of cirrhosis complications in various studies. Pooled data presented in several meta-analyses show that BCAA supplementation significantly increases muscle mass and serum albumin and decreases hospital admissions and cirrhosis-associated complications such as HE and bacterial infections. A meta-analysis including a total of 1297 cirrhotic patients showed that BCAA supplementation can significantly increase muscle mass, serum albumin and BMI, and reduce cirrhosis-related complications, but offered no significant benefits in terms of mortality [59]. A Cochrane meta-analysis of 16 RCTs with a total of 827 participants with cirrhosis showed that BCAA supplementation decreased the incidence of HE but had no effect on mortality or quality of life [60]. Another meta-analysis showed significantly higher serum albumin and lower rates of ascites and edema in the BCAA group but no significant differences in total bilirubin, AST or ALT [18].

Results in our review regarding Child–Pugh score, BTR, TBil and ascites were variable, with some studies reporting improved outcomes in those receiving BCAAs, while others showing no definite benefits. There were no significant improvements in AST levels in the BCAA group in our meta-analysis, suggesting that BCAAs have no mitigating effect on hepatocyte damage. Three studies associated the BCAA group with significant improvements in npRQ, suggesting a beneficial effect on the state of hypercatabolism and fat oxidation. Although BCAAs have been linked with VEGF suppression, reduction in insulin resistance and inhibition of the IGF axis, no significantly different levels of VEGF, FPG, IRI or HOMA-IR were found in our included studies. There was evidence of improved quality of life; however, the body of evidence was not large enough to make definite conclusions.

BCAAs are generally safe without any reported serious side effects [18]. About 10% of patients experience gastrointestinal side effects, such as nausea, vomiting, bloating and diarrhea, which are usually self-resolving but may affect compliance [42,60]. Only one of our included studies mentioned a very low incidence of gastrointestinal side effects related to the supplements [29]. Another potential risk of BCAA supplementation is increased glutamine synthesis. Glutamine is catabolized to ammonia in the intestine and kidneys. In healthy patients, the enhanced ammonia production is negligible, but in patients with liver disease it may be detrimental, leading to HE. Some strategies have been proposed to limit the breakdown of glutamine, including concomitant administration of a-ketoglutarate and/or phenylbutyrate [48].

BCAA supplementation can theoretically decrease the incidence of HE by decreasing ammonia levels and improving the Fischer ratio. Under normal conditions, BCAAs activate the synthesis of alanine and glutamine in skeletal muscle, which are catabolized to ammonia by enterocytes. After they get transferred to the liver via the portal vein, they get detoxified to urea and get excreted in urine. In cirrhosis, due to sarcopenia and BCAA depletion, the detoxifying capacity is impaired. Ammonia escapes detoxification and remains in the bloodstream, resulting in HE. It promotes glutamine synthesis in skeletal muscle, which further enhances ammonia production. Moreover, due to the low Fischer ratio, the increased AAAs cause an imbalance in the neurotransmitter levels of the brain. Hemoglobin released after an episode of gastrointestinal hemorrhage can also trigger HE by means of BCAA antagonism [60]. A meta-analysis of 16 RCTs with a total of 827 participants with cirrhosis showed that BCAA supplementation decreased the incidence of HE but had no effect on mortality or quality of life [60]. In our review, ammonia levels were significantly lower in the BCAA group in two studies, and no difference in the incidence of HE was found in two studies.

Sarcopenia, defined as generalized muscle degradation, is a significant comorbidity in chronic liver disease that may be encountered in up to 70% of cirrhotic patients [61]. Its etiology is multifactorial, including accelerated catabolism, protein imbalances and reduced oral intake [62]. Sarcopenia can be objectively measured by various techniques, such as anthropometry, bioelectrical impedance analysis and computed tomography [63]. Although it can affect patients with liver disease at any age, the elderly population is more at risk because of age-related frailty [64]. As shown by numerous studies, sarcopenia is an important prognostic factor in patients undergoing liver cancer treatment [65]. Sarcopenia has been associated with increased overall mortality and tumor recurrence rates in patients after chemotherapy or resection [66] and with decreased overall survival in patients undergoing locoregional therapies [67,68,69]. BCAA supplementation has been proposed as part of the treatment algorithm for the management of sarcopenia [63]. Only one of the studies included in this systematic review measured sarcopenic indices on its subjects, with no reported benefits. Further studies are needed to identify potential benefits of BCAA supplementation on skeletal muscle mass.

Several controversies have been addressed in the literature with regards to BCAA supplementation. It has been reported that patients with obesity, NAFLD and diabetes mellitus have significantly elevated circulatory BCAA levels. Suggested mechanisms include the inhibition of BCAA catabolism by fatty acid oxidation or the suboptimal rate of BCAA catabolism compared with the accelerated muscle proteolysis [70]. Elevated BCAA levels have been linked to the development of type 2 diabetes in patients with NAFLD, but no causative relationship has been shown [71]. Consumption of BCAA supplements in patients with NAFLD and obesity may increase the liver fat content and further impair glucose metabolism [72]. Some studies suggest that increased BCAA levels may contribute to insulin resistance via the mTORC1 signaling pathway; however, other studies suggest that BCAA supplementation alone is unlikely to impair insulin sensitivity [73]. A study found increased tissue concentration of BCAA within HCC tumors secondary to suppression of catabolic enzymes, such as the rate-limiting branched-chain ketoacid dehydrogenase. The local accumulation of BCAAs was found to activate the mTORC1 pathway, and the degree of catabolic enzyme suppression correlated with tumor growth [74].

Our study comes with several limitations. The number of included studies was relatively small. Most of the included studies were performed in East Asia, and therefore their results may lack generalizability. Moreover, some studies were limited by their retrospective nature, small sample sizes and short follow-up intervals. The majority of the studies did not report on the overall daily caloric and protein intake of their participants, which could potentially overestimate BCAA effects in case of a discrepancy between the two groups. There was also significant heterogeneity in the type, dose and duration of BCAA supplementation among the included studies. Finally, our review only included published studies, which may be subject to publication bias.

## 5. Conclusions

Our systematic review and meta-analysis demonstrated significantly higher post-treatment serum albumin levels in HCC patients undergoing locoregional therapies compared to controls. Although there are studies reporting additional favorable outcomes linked with BCAAs, there are currently not sufficient data to make robust conclusions. Further studies with larger sample sizes and a more diverse patient population need to be conducted to increase the body of evidence. Several topics need to be addressed by future studies, including standardization of supplement dose, type and timing.

## Figures and Tables

**Figure 1 cancers-15-00926-f001:**
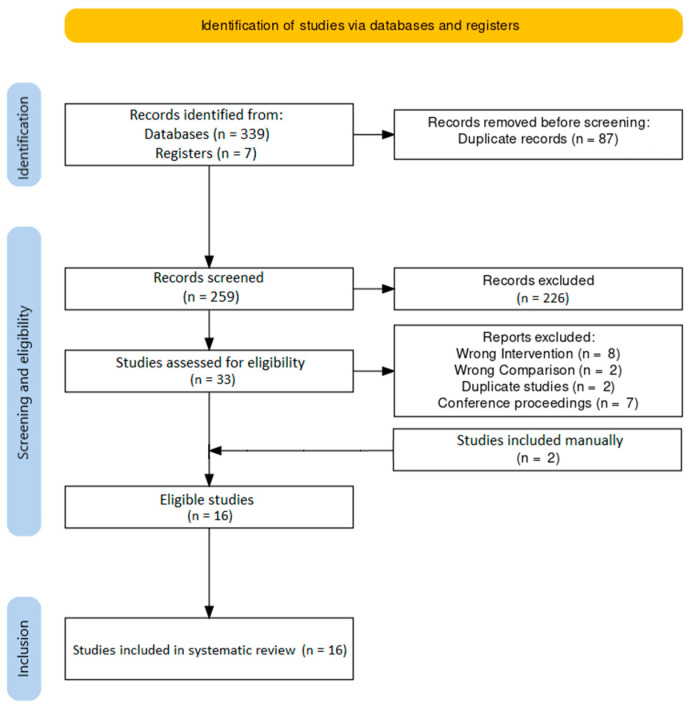
PRISMA flowchart.

**Figure 2 cancers-15-00926-f002:**
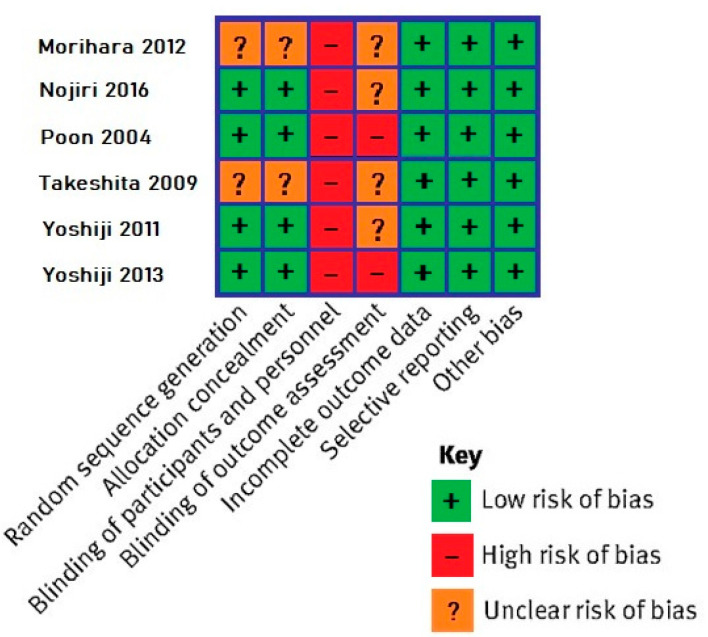
Results of the Cochrane tool for bias assessment in randomized controlled trials [29,32,33,35,37,38].

**Figure 3 cancers-15-00926-f003:**
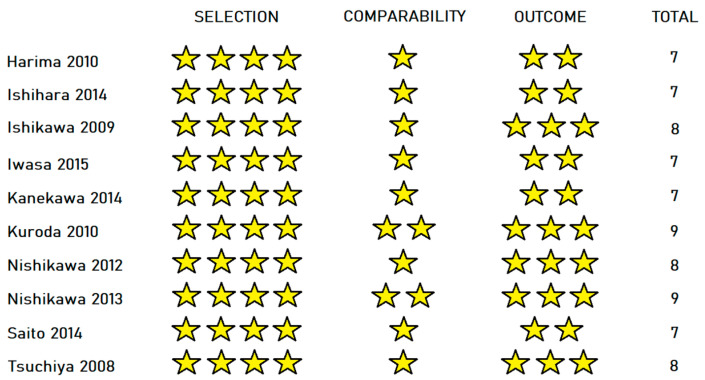
Results of the Newcastle–Ottawa scale for cohort studies [23,24,25,26,27,28,30,31,34,36].

**Figure 4 cancers-15-00926-f004:**
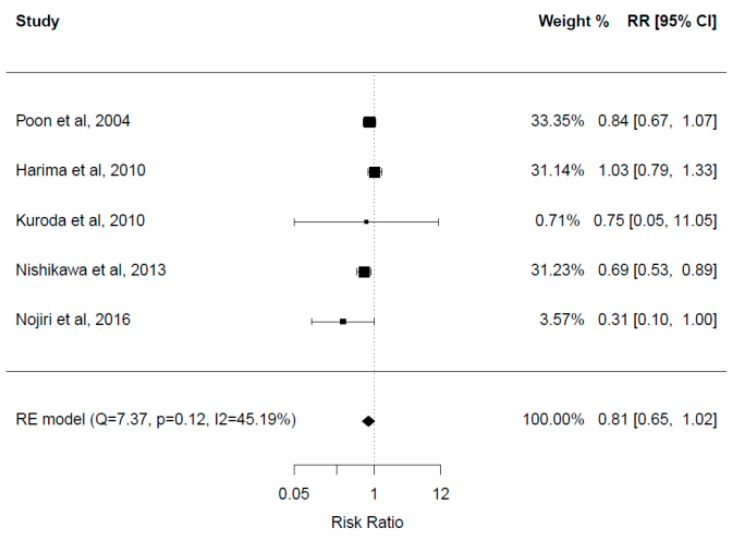
Forest plot for mortality rate [23,28,31,32,33].

**Figure 5 cancers-15-00926-f005:**
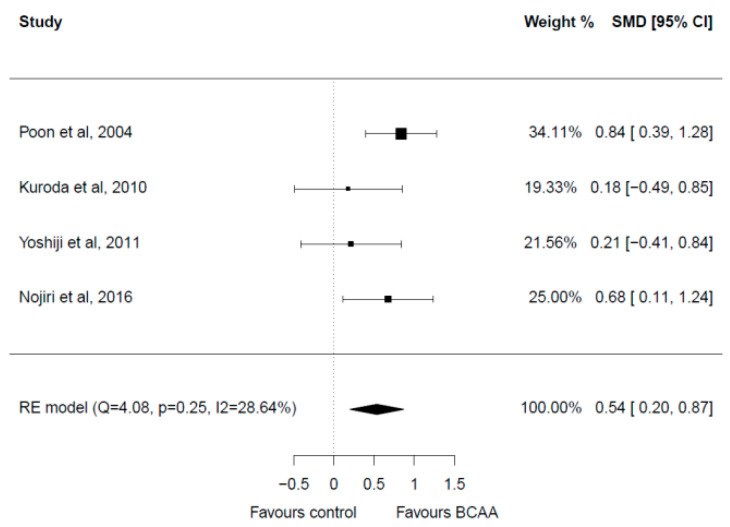
Forest plot for albumin [28,32,33,37].

**Figure 6 cancers-15-00926-f006:**
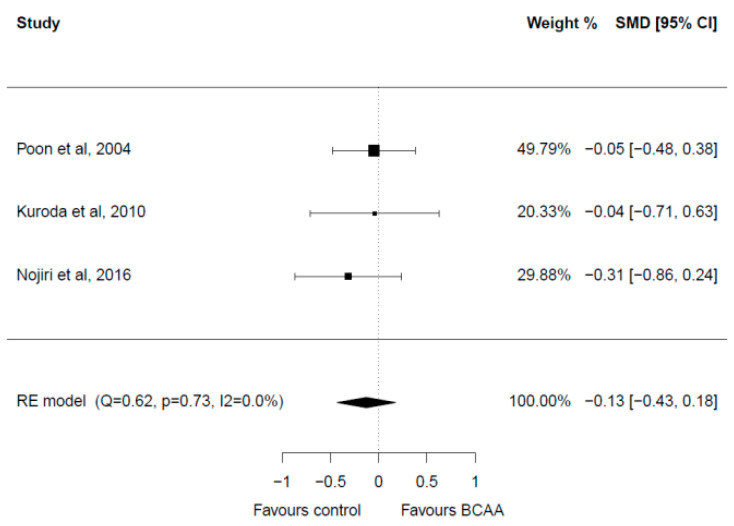
Forest plot for AST [28,32,33].

**Table 1 cancers-15-00926-t001:** List of included studies and baseline demographic characteristics.

Author	Type of Study	Study Period	Group	N	Age (Years)	M/F	HBV/HCV/Other	Stage I/II/III/IV	Child–Pugh A/B/C	Max Tumor Size (cm)	Mean Tumor Number
Harima et al. 2010 [23]	Cohort ^r^	12/2007–2/2009	Control	10	66.4 ± 12.8	8/2	1/8/1	1/2/6/1	6/4/0	NR	NR
			BCAA	13	64.5 ± 9.5	11/2	5/7/1	1/3/3/6	6/7/0	NR	NR
Ishihara et al. 2014 [24]	Cohort ^r^	4/2004–4/2012	Control	86	72.2 ± 8.6	57/29	13/49/24	21/49/11/10	74/11/1	2.9 ± 2.6	NR
			BCAA	76	71.7 ± 7.2	46/30	9/64/3	13/14/21/28	35/36/5	2.9 ± 3.2	NR
			Control	68	71.9 ± 6.5	47/21	7/47/14	21/40/6/2	59/9/0	2.1 ± 0.8	NR
			BCAA	40	73.0 ± 6.0	13/27	0/34/6	16/16/7/1	23/16/1	2.0 ± 0.8	NR
Ishikawa et al. 2009 [25]	Cohort ^r^	5/2002–11/2006	Control	17	71.6 ± 8.3	7/10	NR	NR	0/15/2	2.2 ± 0.7	1
			BCAA	11	68.5 ± 7.4	4/7	NR	NR	0/5/6	2.3 ± 0.6	1
Iwasa et al. 2015 [26]	Cohort ^p^	1/2013–12/2013 & 9/2014–11/2014	Control	84	74.0 ± 8.0	65/18	16/44/22	16/30/21/16	71/12/0	NR	NR
			BCAA	36	70.0 ± 7.0	23/13	6/23/5	10/15/6/5	20/15/1	NR	NR
Kanekawa et al. 2014 [27]	Cohort ^r^	1/2000–12/2011	Control	43	68.0 ± 7.0	34/9	6/30/7	0/0/14/29	30/13/0	NR	NR
			BCAA	49	66.3 ± 7.0	43/6	8/30/11	0/0/8/41	23/26/0	NR	NR
Kuroda et al. 2010 [28]	Cohort ^p^	10/2005–10/2006	Control	15	66.0 ± 8.1	9/6	NR	5/8/2/0	6/8/1	2.0 ± 0.4	1.9 ± 0.6
			BCAA	20	65.6 ± 7.0	13/7	NR	6/11/3/0	8/11/1	2.0 ± 0.6	1.9 ± 0.5
Morihara et al. 2012 [29]	RCT (pilot)	4/2005–6/2006	Control	10	69.3 ± 8.9	7/3	0/10/0	NR	7/3/0	2.4 ± 0.8	1.9 ± 1.2
			BCAA (a.m.)	10	66.9 ± 9.7	8/2	0/9/1	NR	7/3/0	2.4 ± 0.8	1.7 ± 0.9
			BCAA (p.m.)	10	73.5 ± 8.5	8/2	0/9/1	NR	9/1/0	2.0 ± 1.1	1.6 ± 0.8
Nishikawa et al. 2012 [30]	Cohort ^r^	1/2004–1/2010	Control	59	73.2 ± 10.1	32/27	8/43/10	1/11/35/12	39/10/1	3.6 ± 1.5	NR
			BCAA	40	69.9 ± 8.8	27/13	2/28/10	0/12/23/5	22/15/3	3.3 ± 1.7	NR
Nishikawa et al. 2013 [31]	Cohort ^r^	1/2004–10/2011	Control	141	70.9 ± 7.8	83/58	0/141/0	60/60/21/0	88/52/1	2.0 ± 0.7	1.1 ± 0.4
			BCAA	115	69.3 ± 9.4	64/51	115/0/0	41/58/16/0	83/30/2	2.0 ± 0.6	1.1 ± 0.4
Nojiri et al. 2016 [32]	RCT	8/2009–4/2012	Control	26	69.1 ± 11.0	15/11	22/2/2	14/9/3/0	23/3/0	1.8 ± 0.6	1.5 ± 0.5
			BCAA	25	69.7 ± 9.0	15/10	22/1/2	15/8/2/0	21/4/0	1.8 ± 0.6	1.4 ± 0.5
Poon et al. 2004 [33]	RCT	7/1998–12/2000	Control	43	57.9 ± 12.1	39/4	NR	NR	NR	7.1 ± 3.6	NR
			BCAA	41	58.0 ± 13.9	39/2	NR	NR	NR	7.4 ± 3.4	NR
Saito et al. 2014 [34]	Cohort ^p^	8/2009–12/2012	Control	27	70.0 ± 1.9	16/11	5/18/4	14/10/3/0	NR	NR	NR
			BCAA	13	73.4 ± 2.2	8/5	0/11/2	4/7/1/1	NR	NR	NR
Takeshita et al. 2009 [35]	RCT	1/2004–12/2005	Control	28	70.6 ± 9.8	21/7	NR	NR	NR	NR	NR
			BCAA	28	69.1 ± 8.2	19/9	NR	NR	NR	NR	NR
Tsuchiya et al. 2008 [36]	Cohort ^r^	4/1999–9/2004	Control	190	67.2 ± 9.0	127/63	NR	44/113/33/0	NR	2.4 ± 0.9	NR
			BCAA	85	67.8 ± 6.9	41/44	NR	20/41/24/0	NR	2.3 ± 0.9	NR
Yoshiji et al. 2011 [37]	RCT	5/2004–7/2006	Control	26	62.5 ± 11.5	16/10	18/6/8	18/7/1/0	21/5/0	2.1 ± 0.9	NR
			BCAA	16	63.7 ± 10.8	10/6	10/5/5	10/5/1/0	12/4/0	1.8 ± 0.9	NR
Yoshiji et al. 2013 [38]	RCT	NR	Control	42	62.2 ± 14.8	25/17	6/32/13	25/16/1/0	33/9/0	2.2 ± 0.9	1.7 ± 1.2
			BCAA	51	63.6 ± 15.3	32/19	9/36/15	29/20/2/0	41/10/0	2.2 ± 0.9	1.8 ± 1.0

^p^: prospective cohort study. ^r^: retrospective cohort study. RCT: randomized controlled trial. NR: not reported.

**Table 2 cancers-15-00926-t002:** Baseline laboratory characteristics.

Laboratory Value	Controls	BCAA	Significance Level	Studies
Albumin (mg/dL)	3.6 ± 0.2	3.4 ± 0.2	*p* > 0.05	[23,24,26,27,28,29,30,31,32,33,34,35,37,38]
PT (%)	86.2 ± 5.9	82.3 ± 5.1	*p* > 0.05	[23,24,27,28,29,30,31,32,34]
TBil (g/dL)	1.0 ± 0.1	1.1 ± 0.2	*p* > 0.05	[23,24,27,28,29,30,31,32,34,35]
AFP (ng/mL)	3713.3 ± 10,982.3	1872.3 ± 4481.1	*p* > 0.05	[24,26,27,29,30,31,32,34,37,38]
AST (IU/L)	61.3 ± 6.4	64.7 ± 10.4	*p* > 0.05	[28,29,30,31,32,33,34,35]
ALT (IU/L)	56.3 ± 10.0	54.5 ± 12.0	*p* > 0.05	[23,27,29,30,31,32,34,35,37,38]
Ammonia (ug/dL)	52.3 ± 5.3	77.0 ± 20.2	*p* > 0.05	[23,26,27,35]
PLT (×10^4^/mm^3^)	11.7 ± 2.6	10.0 ± 1.2	*p* > 0.05	[27,28,29,30,31,34,35]
BTR	4.2 ± 0.8	3.7 ± 0.3	*p* > 0.05	[23,28,29,32,35]

PT: prothrombin time. Tbil: total bilirubin. AFP: alpha fetoprotein. AST: aspartate aminotransferase. ALT: alanine aminotransferase. PLT: platelet count. BTR: BCAA-to-tyrosine ratio.

**Table 3 cancers-15-00926-t003:** Locoregional treatment and BCAA supplementation characteristics.

Author	Group	N	RFA	TACE	TACE + RFA	HAIC	BCAA Type	BCAA Dose	BCAA Initiation *	Follow-Up Time	Endpoints for Labs
Harima et al. 2010 [23]	Control	10				10				>12 months	5 weeks
	BCAA	13				13	Aminoleban EN	50 g qd (10 p.m.)	NR		
Ishihara et al. 2014 [24]	Control	86		86						10 days	2, 5, 10 days
	BCAA	76		76			LIVACT granules	4.15 g tid	≥2 weeks prior		
	Control	68	68							10 days	2, 5, 10 days
	BCAA	40	40				LIVACT granules	4.15 g tid	≥2 weeks prior		
Ishikawa et al. 2009 [25]	Control	17	17							12 months	1, 6, 12 months
	BCAA	11	11				Aminoleban EN	NR	NR		
Iwasa et al. 2015 [26]	Control	84	22	43	17	2				17 ± 8 days	Time of discharge (17 ± 8 days)
	BCAA	36	12	14	8	2	LIVACT granules or Aminoleban EN	4.15 g tid or 50 g bid respectively	≥4 weeks prior	21 ± 15 days	Time of discharge (21 ± 15 days)
Kanekawa et al. 2014 [27]	Control	43				43				>2 years	At every chemo session (every 5 days × 4 weeks)
	BCAA	49				49	LIVACT granules or Aminoleban EN	4.15 g tid or 50 g bid respectively	Immediately after		
Kuroda et al. 2010 [28]	Control	15	15							12 months	3, 6, 9, 12 months
	BCAA	20	20				Aminoleban EN	50 g bid	NR		
Morihara et al. 2012 [29]	Control	10	10							12 weeks	1, 4, 12 weeks
	BCAA (a.m.)	10	10				Aminoleban EN	50 g qd (7–10 a.m.)	NR		
	BCAA (p.m.)	10	10					50 g qd (10 p.m.)			
Nishikawa et al. 2012 [30]	Control	59		59						6 months	1, 3, 6 months
	BCAA	40		40			LIVACT granules	4.15 g tid	NR		
Nishikawa et al. 2013 [31]	Control	141	141							2.5 ± 1,3 years	NA
	BCAA	115	115				LIVACT granules	4.15 g tid	NR	2.7 ± 1.5 years	
Nojiri et al. 2016 [32]	Control	26			26					60 months	12 months
	BCAA	25			25		Aminoleban EN	50 g bid	2 weeks prior		
Poon et al. 2004 [33]	Control	43		43						30.1 ± 5.7 months	3, 6, 9, 12 months
	BCAA	41		41			Aminoleban EN	50 g bid	1 week prior	29.4 ± 6.0 months	
Saito et al. 2014 [34]	Control	27	27							3 months	7 days, 3 months
	BCAA	13	13				LIVACT granules	4.15 g tid	≥12 weeks prior		
Takeshita et al. 2009 [35]	Control	28		28						2.7 years	2 weeks
	BCAA	28		28			Aminoleban EN	50 g qd (10 p.m.)	1 day prior	2.9 years	
Tsuchiya et al. 2008 [36]	Control	190	190							>2 years	NA
	BCAA	85	85				LIVACT granules	4.15 g tid	NR		
Yoshiji et al. 2011 [37]	Control	26	26							48 months	12 months
	BCAA	16	16				LIVACT granules	4.15 g tid	NR		
Yoshiji et al. 2013 [38]	Control	42	42							60 months	NA
	BCAA	51	51				LIVACT granules	4.15 g tid	NR		

*: with respect to timing of locoregional therapy. RFA: radiofrequency ablation. TACE: transarterial chemoembolization. HAIC: hepatic arterial infusion chemotherapy. EN: enriched nutrient. Qd: once daily. Bid: twice daily. Tid: three times daily. NR: not reported. NA: not applicable.

**Table 4 cancers-15-00926-t004:** Summary of evidence.

Outcome	Total # of Studies	BCAA > Controls	Controls > BCAA	No Difference
OS	6	2 studies [31,32] + 3 studies * [23,27,36]		1 study [35]
Mortality Rate	5		2 studies [30,32]	3 studies [23,28,33]
RFS	2	1 study [31]		1 study [30,32]
Recurrence Rate	6		3 studies [31,32,38]	3 studies [28,36,37]
EFS	2	1 study [32]		1 study [28]
Overall Events, ascites	3		2 studies [32,33]	1 study [28]
Readmission rates, peripheral edema	1		1 study [33]	
Variceal rupture	3			3 studies [28,32,33]
HE, liver abscess	2			2 studies for each **
Liver failure, jaundice, tumor rupture, biloma, intra-abdominal bleeding, renal failure, pneumothorax/hemothorax	1			1 study for each **
Child–Pugh score	5	3 studies [27,29,30]		2 studies [32,34]
Albumin	13	11 studies [23,27,28,29,30,32,33,34]		2 studies [25,37]
AST/ALT	9	1 study [23]		8 studies [24,27,28,32,33,34,35,37]
Ammonia	2		1 study [35] + 1 study * [26]	
TBil	6		2 studies [29,33]	4 studies [27,28,34,35]
PT	6	1 study [34]		5 studies [27,28,29,32,33]
BTR	6	2 studies [23,32]		4 studies [24,26,28,35]
FPG	5			5 studies [23,28,32,35,37]
ChE	3	2 studies [23,35]		1 study [34]
BMI	3			3 studies [23,33,35]
npRQ	3	3 studies [23,28,34]		
Quality of life	3	2 studies (SF-8) [28,32] + 1 study (FACT-G) [33]		
GGT, LDH, HOMA-IR, IRI	2			2 studies for each **
CRP, TProt, TChol, LYMPH, RBC, VEGF, TG	1			1 study for each **

*: significant results only for certain subgroups. See text for further details. **: see text for exact references for each variable. OS: overall survival. RFS: recurrence-free survival. EFS: event-free survival. HE: hepatic encephalopathy. AST: aspartate aminotransferase. ALT: alanine aminotransferase. TBil: total bilirubin. PT: prothrombin time. BTR: branched-chain amino acids-to-tyrosine ratio. FPG: fasting plasma glucose. ChE: cholinesterase. BMI: body mass index. npRQ: non-protein respiratory quotient. GGT: gamma glutamyl transferase. LDH: lactate dehydrogenase. HOMA-IR: homeostatic model assessment for insulin resistance. IRI: immunereactive insulin. CRP: C-reactive protein, TProt: total protein. TChol: total cholesterol. LYMPH: lymphocyte count. RBC: erythrocyte count. VEGF: vascular endothelial growth factor. TG: triglycerides.

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
