# Peer review of "The Role of Branched-Chain Amino Acid Supplementation in Combination with Locoregional Treatments for Hepatocellular Carcinoma: Systematic Review and Meta-Analysis"

_cancers, 2023, doi:10.3390/cancers15030926_

Round 1

Reviewer 1 Report

Thank you for the opportunity to review the manuscript. From this manuscript, the authors analyzed sixteen studies with a 30 total of 1594 patients, and found that BCAA supplementation is associated with higher post-treatment albumin levels. Congratulations to the authors. Interesting article dealing with important topic. Therefore, after reviewing the manuscript carefully, I think this study could be accepted for publication.

Author Response

Thank you very much for your feedback. We have made some minor improvements and have added some new content to the discussion with a few more references.

Reviewer 2 Report

The systematic review "The role of branched-chain amino acid supplementation in combination with locoregional treatments for hepatocellular carcinoma: systematic review and meta-analysis" adds knowledge to the field and presents potentially interesting findings. Nevertheless, some questions should be addressed in order to improve its scientific quality:

- Titles of each subsection in the Results section should be substitute by a sentence stating the results obtained.

- A concluding sentence is needed at the end of each subsection in the Results section.

- The Introduction provides sufficient information to understand the state-of-the-art and citations are appropiate.

-  Only Child-Pugh score is mentioned, what about Maddrey and ABIC?

- In the case of liver cancer, does the staging correspond to the he Barcelona Clinic Liver Cancer (BCLC) Staging System?

- Please, briefly discuss the role of basal diet, obesity, sex and age.

Author Response

Dear reviewer, thank you for your valuable input.

Several spelling/grammar errors have been corrected.

The included papers only mentioned the Child-Pugh score of their included patients. Maddrey and ABIC scores were not provided.

Regarding staging, the staging system that was used by all studies was the one proposed by the Liver cancer study group of Japan, since most of them originated from there. We have added a note of that in the results section.

Regarding the Results section, we do not think that the title of each subsection should be an entire sentence, especially since there are 18 subsections. This would decrease the readability of the paper. We feel that adding a concluding sentence after each subsection would be redundant, that is why we have provided a summarizing table instead. 

A new paragraph has been added to the discussion which touches on sarcopenia. A paragraph discussing obesity was already present in the discussion and we believe is adequate. A comment about daily caloric intake has been added in the results section, as well as in the limitations. The disparities of gender in liver disease are complex and we do not believe they are immediately pertinent to the topic of this paper, especially since none of our included studies discussed this issue.